# Radiotherapy Upregulates the Expression of Membrane-Bound Negative Complement Regulator Proteins on Tumor Cells and Limits Complement-Mediated Tumor Cell Lysis

**DOI:** 10.3390/cancers17142383

**Published:** 2025-07-18

**Authors:** Yingying Liang, Lixin Mai, Jonathan M. Schneeweiss, Ramon Lopez Perez, Michael Kirschfink, Peter E. Huber

**Affiliations:** 1Molecular and Radiation Oncology, German Cancer Research Center (DKFZ), 280 INF, 69120 Heidelberg, Germany; liangyingying1984@126.com (Y.L.); lixin.mai@dkfz-heidelberg.de (L.M.); jonathan.schneeweiss@dkfz-heidelberg.de (J.M.S.); r.lopez@dkfz-heidelberg.de (R.L.P.); 2Institute of Immunology, University of Heidelberg, Im Neuenheimer Feld 305, 69120 Heidelberg, Germany; 3Radiation Oncology, University Hospital Heidelberg, 400 INF, 69120 Heidelberg, Germany

**Keywords:** complement activation, radiotherapy, membrane-bound complement regulatory proteins (mCRPs), CD20, complement-dependent cytotoxicity (CDC), timing, DNA-damage repair, radiosensitivity

## Abstract

Combining two main columns of cancer therapy, such as radiotherapy and immunotherapy, yields high potential for improved treatment options for cancer patients. In this study, we aimed to explore the interplay between radiotherapy and the complement system, which represents an important part of the human innate immune system on a basic research level. Our findings here will contribute to a deeper understanding of the interaction between the human innate immune system and radiotherapy and will help to find new therapeutic ways to prime the human innate immune system to treat cancer together with radiotherapy.

## 1. Introduction

Radiotherapy (RT) is used for more than 50% of all cancer patients. In particular physical advancements and precise tumor targeting have resulted in fewer adverse effects and improved tumor cure [1]. Ionizing radiation induces DNA damage, specifically DNA double-strand breaks (DSBs), leading to various forms of cellular death, including apoptosis, necrosis, mitotic catastrophe, and ferroptosis [2]. Many of these processes are interdependent with radiation-induced immune responses [3,4], and therapeutic resistance to radiotherapy may develop through immune escape mechanisms related to these processes [5,6]. This may lead to cancer treatment failure, metastasis, cancer recurrence, enhanced radiation toxicity [7,8], and a poor prognosis [9].

Immunotherapy, such as immune checkpoint blockade (ICB), therapy has emerged as a significant treatment modality for various cancers, but its efficacy in combination with RT remains unclear [10,11]. Much less is known about the role of the complement system in the response of tumors to radiotherapy, although the complement system and its regulation are a pivotal effector mechanism in the innate immune response to tumors [12,13,14,15]. Moreover, complement activation plays an important role in antibody-based immunotherapy, which has become an important clinical cancer treatment modality. Three distinct pathways have been elucidated in the cascades of complement activation. The classical pathway is dependent on the antigen–antibody immune complexes [16]. The lectin pathway of the complement system initiates when mannose-binding lectin (MBL) binds sugar residues on cell surfaces, leading to the activation of mannose-associated serine proteases (MASPs) [17]. The alternative pathway relies on the spontaneous hydrolysis of C3 in the blood, which is at a consistently low level. Ultimately, all of these pathways share a chain of final steps utilizing C3 and C5 convertases to form the membrane attack complex (C5b-9, MAC). Furthermore, complement activation also leads to two other main effector mechanisms: the opsonization of target cells by iC3b/C3b and the recruitment of immune cells via anaphylatoxins C3a and C5a [18].

Antibody-based immunotherapy has become an important treatment strategy for various malignancies [19]. The efficacy of monoclonal antibodies (mAbs) relies on multiple immune effector mechanisms: (a) complement-dependent cytotoxicity (CDC), (b) antibody-dependent cell-mediated cytotoxicity (ADCC), (c) direct induction of apoptosis by targeting certain molecules [20], and (d) antibody-dependent cellular phagocytosis (ADCP) [21,22,23]. The resistance of tumor cells to antibody-based cancer therapy mediated by CDC is primarily attributable to the overexpression of membrane-bound complement regulatory proteins (mCRPs), CD46, CD55, and CD59 [24,25], which counteract complement activation. CD46 (membrane cofactor protein, MCP) aids the cleavage of C3b and C4b mediated by factor I, CD55 (decay-accelerating factor, DAF) accelerates the decay of C3 and C5 convertases, and CD59 (protectin) prevents the assembly of the MAC [26].

To date, the relationship between radiotherapy, tumors, and the complement system is less well understood and remains controversial [12,27]. Elvington et al. [28] demonstrated that a combinatorial strategy of radiotherapy with complement inhibition resulted in improved anti-tumor effects in a subcutaneous murine lymphoma model. Conversely, complement activation with the generation of C3a and C5a has been reported to be essential for the success of radiotherapy in models of murine melanoma, colon cancers, and human skin cancers [29].

Here, we investigate the interaction of complement-based anti-cancer therapy with radiation. We find that radiation markedly and robustly upregulates mCRPs across a spectrum of hematological and solid cancer cells. We also provide evidence that complement activation followed by radiation enhances the efficacy of complement-dependent cytotoxicity (CDC), thus inducing stronger anti-tumor effects.

## 2. Materials and Methods

### 2.1. Cell Culture

All the cell lines were purchased from the American Type Culture Collection (ATCC, Manassas, Virginia, USA). Human prostate cancer cell line Du145 (Cat. No. #HTB-81), human lung adenocarcinoma cell line A549 (#CCL-185), Burkitt lymphoma cell lines Raji (#CCL-86) and Ramos (#CRL-1596), human acute lymphocytic leukemia (ALL) cell line Reh (#CRL-8286), human chronic myeloid leukemia (CML) cell line K562 (#CRL-3344), human hepatocellular carcinoma cell line HepG2 (#HB-8065), and human breast cancer cell line MCF-7 (#HTB-22) were cultured in the RPMI 1640 or DMEM medium (Life Technologies, Darmstadt, Germany) with L-glutamine (PAA, Freiburg im Breisgau, Germany), supplemented with 10% heat-inactivated fetal calf serum (FCS) (Invitrogen, Darmstadt, Germany), and maintained at 37 °C and 5% CO_2_.

### 2.2. The ^51^Chromium Release Assay

To determine the effect of radiation on complement activation, a ^51^Chromium release assay was conducted to assess the complement-mediated cell lysis. Tumor cells were irradiated with 2 Gy or 8 Gy versus a 0 Gy control. A total of 72 h after radiation, target cells (1 × 10^5^/20 μL full medium) were collected and labeled with 10 μCi ^51^Cr (Hartmann Analytic, Braunschweig, Germany) by incubation for 1.5 h at 37 °C. Cells were then washed in an assay medium (0.6 mM MgCl_2_, 0.1% BSA in RPMI or DMEM without FCS) and re-suspended in an assay medium. A total of 10^4^ cells/50 μL were seeded in triplicate into the 96-well plate and incubated with rabbit anti-Du145/anti-K562 polyclonal antibodies (Abs, 1:50 dilution) (as described previously [30]) or anti-CD20 monoclonal Ab, Rituximab (Roche Diagnostics GmbH, Penzberg, Germany), for 30 min at 37 °C. Normal human serum (NHS, diluted in a 1:5 assay medium) added as the complement source or heat-inactivated NHS (NHS i.a.) as the negative control were incubated for 1 h at 37 °C. NHS was prepared from a pool of sera of ten healthy donors and stored frozen at −70 °C, while NHS i.a. was generated by heating the sera to 56 °C for 30 min. Maximal ^51^Cr release was determined by the addition of 1% TritonX-100 (Roche, Germany), while the spontaneous ^51^Cr release was measured by the replacement of NHS with an assay medium. Each sample contained spontaneous release, maximal release, complement activation-induced release, and a negative control. Finally, cells were centrifuged at 1200 rpm for 5 min, and the supernatant was collected and measured for radioactivity in a γ-counter (Wizard2 2470, PerkinElmer, Shelton, CT, USA). The percentage of the specific lysis in each sample was calculated automatically according to the following formula: [(test release-spontaneous release)/(maximum release-spontaneous release)] × 100. All experiments were performed in triplicate.

### 2.3. Measurements of Membrane Complement Regulatory Proteins (mCRPs) and CD20

To investigate how radiation influences cell surface complement regulation, mCRPs were measured by flow cytometry. A monoclonal mouse anti-human CD46 antibody (IgG1, clone GB24, 2.49 mg/mL) was kindly provided by Dr. J. Atkinson from Washington University. Mouse anti-human CD55 (IgG1, clone Bric 110, 0.5 mg/mL) and mouse anti-human CD59 (IgG2b, clone Bric 229, 1.4 mg/mL) were purchased from the International Blood Group Reference Laboratory (Birmingham, UK). Cells were irradiated with photon irradiation 2 Gy, 4 Gy, 6 Gy, and 8 Gy, collected after 24 h, 48 h, or 72 h, respectively, and suspended in a FACS buffer (1% BSA, 0.1% NaN3 in PBS). A total of 1 × 10^4^ cells in a 100 μL FACS buffer were seeded in a 96-well plate and incubated with primary antibodies or isotype antibodies at a concentration of 10 μg/mL for 30 min. Cells were washed twice with a FACS buffer and incubated with a FITC-conjugated goat anti-mouse secondary antibody (DAKO, Denmark) in the dark for 30 min at 4 °C. Finally, cells were washed with a FACS buffer and fixed in 100 μL 1% paraformaldehyde in PBS. A measurement was performed with an LSRII flow cytometer and was analyzed using FACSDiva software (Version 7.0; BD Biosciences, Heidelberg, Germany) and FlowJo™ software (version 10.8.1; FlowJo, LLC, Ashland, OR, USA). CD20 expression was measured on lymphoma cells following primary antibody incubation with Rituximab (10 μg/mL) and a secondary antibody. Flow cytometry data were analyzed following the exclusion of debris based on forward and side scatter (FSC vs. SSC) parameters and the subsequent removal of cell doublets using forward scatter area versus height (FSC-A vs. FSC-H) gating.

### 2.4. Clonogenic Survival Assay

For adherent cell lines, cells were incubated with complement-activating antibodies for 30 min, followed by the addition of 1:10 NHS for 1.5 h. Alternatively, cells were irradiated first and incubated with antibodies and NHS. Subsequently, cells were plated in T25 culture flasks (Thermo Scientific, Copenhagen, Denmark), irradiated with 2 Gy, 4 Gy, 6 Gy, or 8 Gy versus a 0 Gy control, and cultured at 37 °C, 5% CO_2_ for 7 days. Afterwards, the colonies (consisting of at least 50 cells) were fixed, stained with a 0.001% crystal violet solution (Serva, Heidelberg, Germany), and the number of colonies was counted manually.

For non-adherent cell lines, cells were plated into 96-well plates and irradiated or treated with Ab and NHS. Ten days later, wells containing apparent cell pellets with a yellow-colored medium were defined as formed colonies.

The surviving fraction of cells was calculated with the following formula: (number of colonies/number of plated cells in the combined treatment group)/(number of colonies/number of plated cells in the RT-only group). Clonogenic survival fraction curves were fitted according to the linear-quadratic model. Data are derived from two to three biological replicates for each cell line and radiation modality.

### 2.5. Apoptosis Measurement

Cells were incubated with complement-activating antibodies and 1:10 NHS, irradiated with 8 Gy either before or after the addition of the corresponding antibody, and incubated at 37 °C, 5% CO_2_ for 72 h. Afterwards, cells were detached with Accutase (PAA Laboratories, Pasching, Austria); the respective medium was collected and centrifuged together at 1200 rpm for 5 min. Cell pellets were washed in PBS, re-suspended in binding buffer FITC (Annexin V apoptosis detection kit, BD Pharmingen, Franklin Lakes, NJ, USA), and labeled with FITC Annexin V (1:20) and Propidium Iodide (PI, 1:20). Finally, samples were analyzed by flow cytometry. Flow-cytometric data were analyzed after debris was excluded by gating on forward versus side scatter (FSC/SSC) plots. Early apoptotic cells were identified as annexin V–FITC-positive/propidium iodide-negative (FITC^+^ PI^−^).

### 2.6. Analysis of Double-Strand Breaks (DSBs) and Cell Cycle Kinetics

Tumor cells were pretreated either with an isotype antibody or the combination of mCRP-specific non-complement activating neutralizing antibodies, anti-CD46 (clone GB-24), anti-CD55 (clone Bric 110), and anti-CD59 (clone Bric 229). A total of 72 h later, cells were irradiated with 4 Gy and collected after 24 h. Samples were subsequently washed, fixed with 3% PFA, permeabilized with 70% ethanol, and incubated with an Alexa Fluor 488 conjugated anti-γH2AX-phosphorylated Ser139 antibody (BioLegend, San Diego, CA, USA) diluted 1:20 in 0.5% BSA/PBS at room temperature for 1 h. Cells were then washed and re-suspended in 1 µg/mL DAPI/PBS. Sample acquisition was performed with an LSR Fortessa flow cytometer. Events from debris and cell doublets/aggregates were excluded. Two independent experiments were performed with each cell line in triplicate, and 10,000–20,000 singlets were acquired from each sample. The sample median fluorescence intensity of γH2AX was used for statistical analysis.

### 2.7. Statistical Analysis

Data are presented as means ± the standard error of the mean (SEM) unless stated otherwise. Statistical analysis was performed using GraphPad Prism (version 10.2.3), applying the unpaired two-tailed *t*-test with 95% confidence interval or, when required, one-way ANOVA with post hoc Dunnett’s multiple comparisons testing. A *p*-value < 0.05 was considered statistically significant.

## 3. Results

### 3.1. Irradiation Upregulates Expression of Membrane-Bound Complement Regulator Proteins (mCRPs) in a Dose- and Time-Dependent Manner

To study possible interactions between complement regulation and irradiation, we assessed whether irradiation affects the expression of the main mCRPs, such as CD46, CD55, and CD59. We irradiated a panel of cancer cell lines of tumor entities where radiotherapy (RT) plays a role in their clinical treatment regimens. The panel contained cell lines derived from lymphoma (Raji, Ramos), leukemia (Reh, K562), breast cancer (MCF-7), lung cancer (A549), prostate cancer (Du145), and liver cancer (HepG2) cells. In most of the cell types, we observed a radiation dose-dependent upregulation of the surface expression of one or more of the mCRPs, CD46, CD56, or CD59 (Figure 1). This upregulation persisted or increased from 48 h up to 72 h after irradiation (Figure 1). At the early time point 24 h post-RT, the solid breast (MCF-7) and lung (A549) cancer cells showed a trend only, while the hematological cancer cells all reached statistical significance, suggesting that the hematological tumor cells at the chosen conditions were more sensitive to RT-induced mCRP upregulation (Appendix A).

### 3.2. Irradiation Induces Upregulation of CD20 Expression on Lymphoma Cells

CD20 is widely expressed on a variety of B-cell lymphoma malignant cell surfaces. This is clinically utilized for lymphoma treatment with CD20-targeting monoclonal antibody drugs such as Rituximab [31]. To test whether radiation impacts the expression of CD20 on lymphoma cells, we irradiated the lymphoma cell lines Ramos and Raji (B-cell lymphoma cell lines) and the leukemic cell line Reh.

On all investigated hematological cell lines, irradiation induced an upregulation in CD20 surface expression, most pronounced at 48 h after RT (Figure 2 and Appendix A). This upregulation remained stable for at least 3 days after irradiation. For doses up to 6 Gy, we could observe a dose-dependent induction reaching a plateau between 48 h and 72 h after RT. Notably, low-baseline CD20-expressing Reh cells showed a distinct relative induction of CD20 expression after RT compared to unirradiated control cells (Figure 2C).

Hence, irradiation triggers upregulation of CD20 expression in hematological tumors as an important target of monoclonal antibody-based immunotherapy.

### 3.3. Radiation Reduces Tumor Cell Lysis After Antibody-Mediated Complement Activation

To functionally assess whether the observed RT-induced upregulation of mCRPs leads to reduced complement-mediated cell lysis, we performed 51Chromium release cytotoxicity assays. Indeed, irradiation of the tumor cells before the treatment of the complement decreased complement-mediated cytotoxicity in solid and hematological tumor cells (Figure 3A,B). In A549 cells, 8 Gy reduced cell lysis to approximately 38% compared with 78% specific cell lysis by complement-activation alone. This effect was even more prominent in hematological tumor entities; for example, in K562 cells, 2 Gy reduced tumor cell lysis to 43% versus 75% without irradiation, and 8 Gy further decreased the lysis to around 4%. Similar results were observed in HepG2 cells (Figure 3D). In contrast, in Du145 prostate cancer cells, radiation did not statistically significantly reduce complement-mediated tumor cell lysis (Figure 3C). It is also worth noting that the specific tumor lysis relied on the participation of the complement by adding NHS among all tested cell lines.

### 3.4. Sequence of RT and Complement Activation Impacts RT-Induced Tumor Cell Apoptosis

We next investigated how targeted pre-activation of the complement impacts RT-mediated induction of apoptosis in tumor cells. In all investigated cell lines, complement activation, induced by anti-tumor antibodies prior to RT, increased early apoptosis rates compared to RT alone (Figure 4). In solid cancer cell lines MCF-7 and HepG2, this effect was most pronounced, leading to doubling (MCF-7 cells, Figure 4A) or tripling (HepG2 cells, Figure 4B) apoptosis rates compared to RT alone, followed by Raji cells. In the other hematological cell lines, K562 and Ramos, this effect was less pronounced without reaching statistical significance (Figure 4C–E). Interestingly, in all analyzed tumor cell lines, apoptosis rates were not affected when tumor cells were exposed to an antibody and complement after RT compared to RT alone.

### 3.5. Intrinsic Radiosensitivity of Tumor Cells Is Not Markedly Altered by Complement Activation or mCRP Neutralization

We also analyzed the effects of complement activation on intrinsic radiosensitivity. As expected, radiation reduced the survival fraction in colony-forming assays of the tumor cell lines in a dose-dependent manner according to the linear-quadratic model. Additional treatment by antibody-based complement activation did not significantly alter cell survival, suggesting that intrinsic radiosensitivity remained largely unchanged (Appendix A). We then neutralized mCRPs and performed flow cytometry-based γH2AX measurements as a surrogate of radiation-induced DNA damage and repair dynamics. Likewise, blocking mCRPs did not alter RT-induced DNA damage repair kinetics in all but one tumor cell line (Appendix A): A549 cells showed slightly increased RT-induced DNA-double strand breaks after mCRP blockade at 1 h after RT, which was diminished at 24 h after irradiation. Taken together, our data indicate that complement activation does not relevantly alter the intrinsic radiosensitivity of tumor cell itself or the DNA-damage processes after radiation.

## 4. Discussion

Immune therapy for cancer using immune checkpoint blocker (ICB) therapy has become an important modality for many tumor entities [32,33]. However, it remains unclear whether combining radiotherapy (RT) with the immune checkpoint blockade (ICB) yields improved outcomes for the majority of tumors compared to either modality alone [10]. Thus, further research to enable the immune system for radiotherapy combinations is important. Here we investigated the interaction of radiotherapy with the complement system, a major component of innate immune responses, as an alternative modality to alter the immune response towards RT.

We found that radiotherapy profoundly impacted the susceptibility of cancer cells to the complement system. Complement activation on cells is effectively controlled by membrane-bound complement regulatory proteins (mCRPs). Our data show that RT dose- and time-dependency induces the upregulation of mCRP on tumor cells, indicating that radiotherapy hampers complement-induced tumor cell killing, which may potentially contribute to radiotherapy resistance [28,29].

We observed the mCRP overexpression across a wide spectrum of solid and hematological tumor cells, suggesting that this effect is largely tumor entity independent. To strengthen the translational relevance, we focused on human cell lines of tumor entities where RT plays a role in their current therapy regimens, such as human A549 non-small cell lung carcinoma (NSCLC) [34], Du145 prostate adenocarcinoma [35], or Raji and Ramos B-cell lymphoma cancer cell lines [36]. Certain differences were observed among cell types with respect to time response dynamics and efficacy, reflecting cell-specific biological heterogeneity, implying tumor cell-specific mechanisms to evade the complement-mediated immune surveillance [37].

This upregulation of mCRPs by RT suggests a potential response mechanism by which RT might self-limit its therapeutic efficacy via the suppression of complement attacks targeting tumor cells. Indeed, we observed that in both solid and hematological cancer cells, specific cell lysis was reduced when additional RT was applied before antibody-mediated complement activation. Previous studies observed a cytokine-mediated upregulation of CD55 and CD59 expression in human hepatoma cells in tumors mediated via pro-inflammatory cytokines such as tumor necrosis factor-alpha, IL-1 beta, and IL-6 [38]. Since RT promotes inflammation in cancer by inducing an upregulation of proinflammatory cytokine expression in cancer cells, among others [39], RT-induced cytokine induction leading to mCRP upregulation might be another potential mechanism to explore further. Thus, more studies on investigating the interaction between the complement system, radiotherapy, and cytokine production are required. In addition to mCRP upregulation, these results may be due to the generation of RT-induced, complement-derived anti-tumor-efficacy-limiting byproducts such as C5a [19,40,41]. Although tumor-directed complement activity is considered anti-tumorigenic (by its cytotoxic and opsonic action), the anaphylatoxin C5a has been shown to also exert a pro-tumorigenic effect in the local tumor microenvironment by recruiting pro-tumorigenic myeloid-derived suppressor cells among others [15,19]. In this context, Elvington et al. [28] reported that combining radiotherapy and tumor-targeted-complement inhibition resulted in longer survival time and tumor control in subcutaneous lymphoma mouse models. The combination enhanced apoptosis in the irradiated area, subsequently increasing neutrophil influx and promoting effective T-cell immunity [28]. Data from other studies support the notion that mechanisms that limit complement activation could improve the anti-tumor effect of radiotherapy, although in contrast, Surace et al. found that interfering with the anaphylatoxins C3a and C5a reduced the efficacy of radiotherapy in murine melanoma, colon cancer, and human skin cancer models indicating that a systemic functionally active complement system yields high anti-tumor potential [29].

The sequencing is likely important for treatment success for the combination of RT and tumor-directed complement activation. We therefore investigated whether complement activation prior to radiotherapy would alter the efficacy of tumor cell killing. We found that tumor-directed complement activation prior to RT was more effective than complement activation after RT, particularly with regard to tumor cell apoptosis. This supports the notion that RT-induced mCRP upregulation on tumor cells self-limits its therapeutic efficacy because it may counteract subsequent complement-mediated tumor cell killing. Vice versa, if the complement system is activated before the application of RT, the combined effect [29] could elicit a chain of cellular apoptosis and other forms of radiation-induced cell death outcomes [2], which may be due to C3b/iC3b fragments [42] and sub-lytic membrane-attack complex (MAC) [43] deposition on the tumor-cell surface while the plasma membrane was still intact without being destroyed by the ionizing RT. This “priming” event by the prior complement activation may lower the threshold of apoptosis shown in Figure 3, by maintaining the docking sites to elicit the following cell death events. Furthermore, via the action of anaphylatoxins, the complement system may also lead to an increased anti-tumor effect of immune cells, such as CD8+ T lymphocytes, suggesting that the complement system is a central mediator of radiotherapy-induced tumor-specific immunity and clinical response [29]. This finding also brings clinical sequencing practicality. Among current RT–immunotherapy protocols, such as RT+ anti-PD-1 (Programmed Cell Death Protein 1)/PD-L1 (Programmed Death-Ligand 1), RT + anti-CTLA-4 (Cytotoxic T-Lymphocyte-Associated Protein 4), and RT + STING (stimulator of interferon genes) agonist, RT is usually followed by immunotherapy or concurrently given with immunotherapy to harness the human immune system after RT-induced antigen release or double-strand breaks, such as the PACIFIC series in NSCLC (or Adriatic in SCLC) where Durvalumab (PD-L1 inhibitor) after concurrent chemoradiotherapy (cCRT) has evolved as the standard of care for patients with unresectable, stage III non-small-cell lung cancer (NSCLC) [44]. However, our findings in this study show that RT should be performed after complement-activation-based immunotherapy because RT could increase the expression of radio-resistant markers, mCRPs, and reduce the treatment efficiency of antibody-based immunotherapy. Meanwhile, this strategy may also reduce the severity of RT-enhanced vascular leakage, which improves treatment safety, offering promising clinical translation opportunities.

Mechanistically, the induction and repair dynamics of DNA damage, including double-strand breaks (DSBs), in cells are the major processes that determine how RT affects tumor cells and are thus parameters of intrinsic radiosensitivity. We found that neither complement activation nor the mCRP blockade (also improving complement effects) had a relevant influence on RT-induced DNA damage, repair kinetics, or intrinsic tumor cell radiosensitivity. The observed anti-tumor effects of combined RT and complement-mediated immunotherapy thus appear to be of another nature. While we found that RT could bolster tumors against immunological complement attacks, the RT-induced modulation of immunologically relevant cell surface markers may also offer the potential for improving tumor responses toward immunotherapy. Of note, in our study, RT enhanced the surface expression of CD20 on hematological cell lines, thus enhancing the targets of CD20 monoclonal antibody immunotherapy may be especially relevant for lymphoma treatment.

## 5. Conclusions

Complement activation is indispensable for immune surveillance by integrating the innate and adaptive immune systems. Our data clearly indicate that tumor-directed complement activation is blocked by overexpression of membrane complement regulators upon prior radiotherapy. This extends our current knowledge on the complexity of mechanisms in the interaction of complement- and radiation-induced anti-cancer immune responses, which suggests a potentially beneficial role for the combination in cancer therapy.

## Figures and Tables

**Figure 1 cancers-17-02383-f001:**
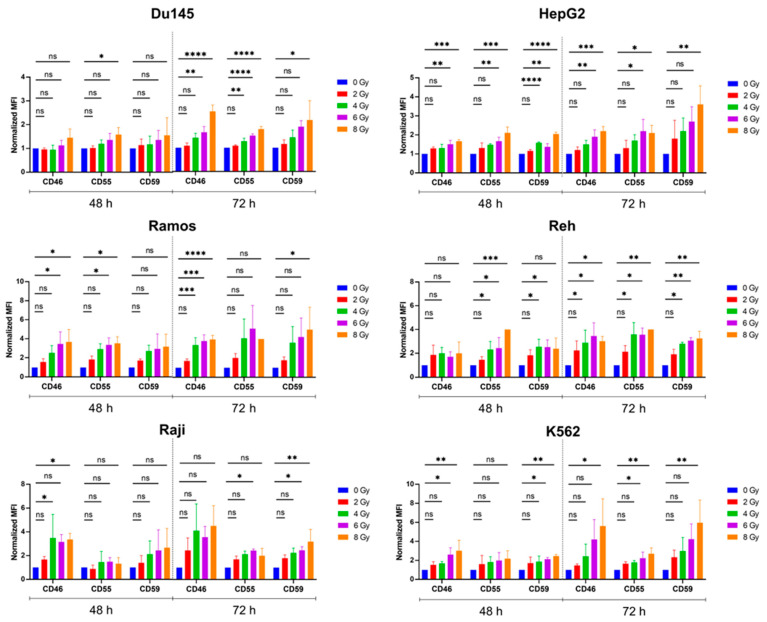
Radiation upregulates the expression of membranous complement regulatory proteins (mCRPs) on different human tumor cells. Different cancer cell lines were treated with different radiation doses as indicated, and the median fluorescence intensity (MFI) was analyzed by flow cytometry 48 h and 72 h after irradiation. Data was analyzed by two-tailed Student’s *t*-test; the MFI was calculated by subtraction from the MFI in the corresponding isotype group; the normalized MFI was calculated by dividing by the values of the 0 Gy group. Data is given as mean values ± the SD of 2–3 independent experiments per cell line; ns, not significant; * *p* < 0.05; ** *p* < 0.01; *** *p* < 0.001; **** *p* < 0.0001 (one-way ANOVA with post hoc Dunnett’s multiple comparisons testing).

**Figure 2 cancers-17-02383-f002:**
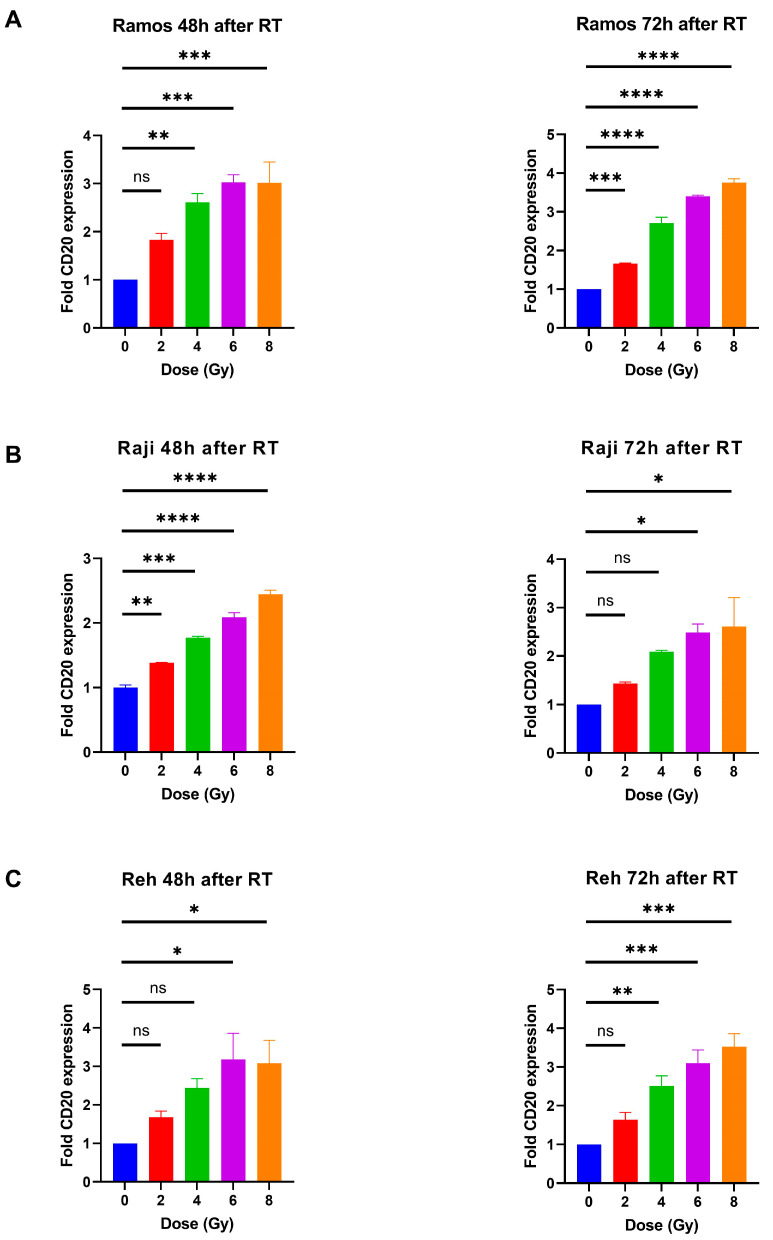
Radiation induces upregulation of CD20 expression in different lymphoma and leukemic cell lines. The lymphoma cell lines Ramos (**A**) and Raji (**B**) and leukemic cell line Reh (**C**) were irradiated with different doses, and 48 h and 72 h after irradiation, CD20 expression was obtained via flow cytometry; ns, not significant; * *p* < 0.05; ** *p* < 0.01; *** *p* < 0.001; **** *p* < 0.0001 (one-way ANOVA with post hoc Dunnett’s multiple comparisons testing).

**Figure 3 cancers-17-02383-f003:**
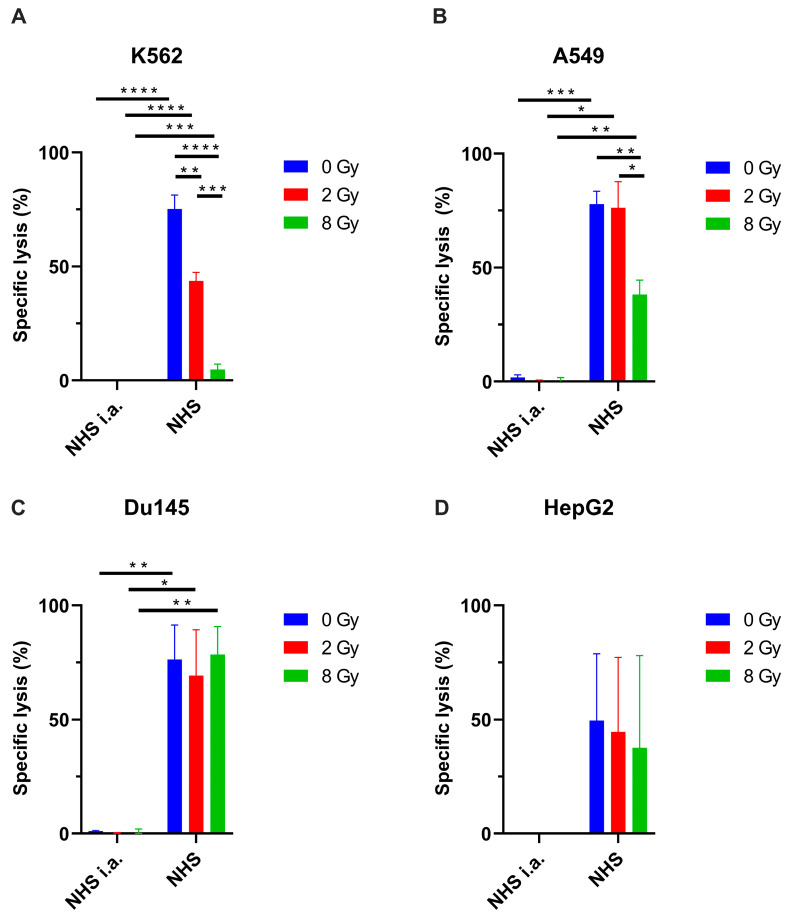
Radiation reduces antibody-induced complement-mediated tumor cell lysis in certain cancer cell lines. (**A**–**C**) Antibody-based complement-mediated cell lysis was observed in all tested cancer cell lines except the hepatic cancer cell line HepG2 (**D**). (**A**,**B**) K562 and A549 cell lines showed that radiation followed by normal human serum treatment reduced the tumor cell lysis. Different cell lines were treated with 0 Gy, 2 Gy, or 8 Gy, and 72 h after radiation, incubated with the corresponding complement-activating antibodies, followed by the addition of normal human serum (NHS) or heat-inactivated NHS (NHS i.a.) as a control. Specific lysis was analyzed by a 51Chromium release assay. Data was analyzed by two-tailed unpaired Student’s *t*-test and given as mean ± SEM values of 2–3 independent experiments per cell line. Only statistically significant findings were indicated; * *p* < 0.05; ** *p* < 0.01; *** *p* < 0.001; **** *p* < 0.0001. NHS: normal human serum; NHS i.a.: inactivated normal human serum.

**Figure 4 cancers-17-02383-f004:**
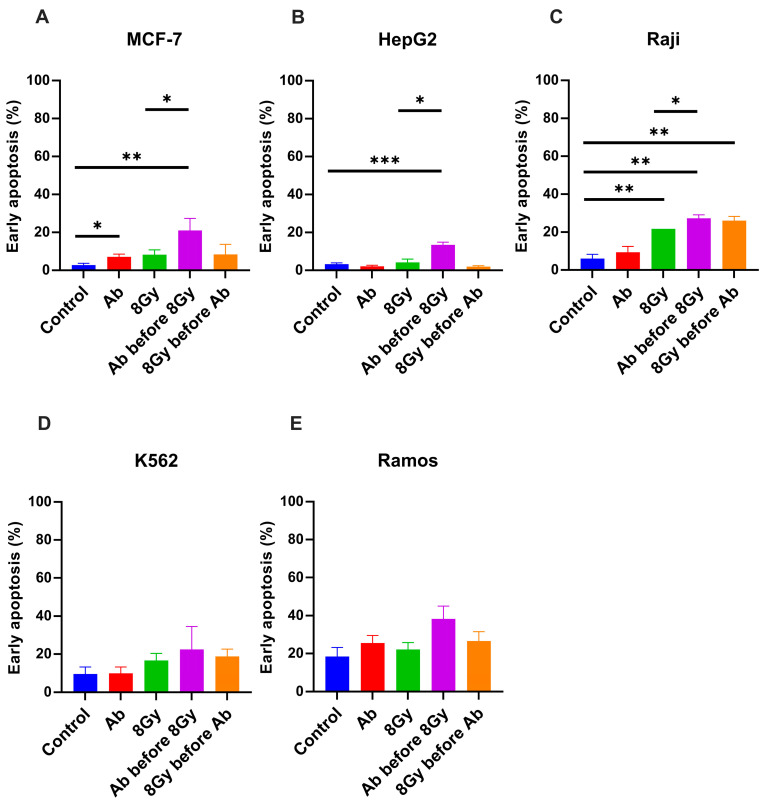
Complement activation before irradiation shows maximal efficacy in inducing early apoptosis. Different cancer cell lines (**A**–**E**) were treated with a complement-activating antibody plus human serum and/or radiation with 8 Gy as indicated, and the percentages of early apoptosis were analyzed by flow cytometry. Data was analyzed by two-tailed unpaired Student’s *t*-test; normalized apoptosis percentages were calculated by dividing by the values of the control group. Data is given as the mean ± SEM of 2–4 independent experiments per cell line. Only statistically significant findings were indicated; * *p* < 0.05; ** *p* < 0.01; *** *p* < 0.001.

## Data Availability

The raw data supporting the conclusions of this article will be made available by the authors upon request.

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
