# Peer review of "Radiotherapy Upregulates the Expression of Membrane-Bound Negative Complement Regulator Proteins on Tumor Cells and Limits Complement-Mediated Tumor Cell Lysis"

_cancers, 2025, doi:10.3390/cancers17142383_

Round 1

Reviewer 1 Report

Comments and Suggestions for Authors

The manuscript “Radiotherapy upregulates the expression of membrane-bound negative complement regulator proteins on tumour cells and limits complement-mediated tumor cell lysis” authored by Liang et al is interesting as it focuses on relationship between radiotherapy complement-dependent cytotoxicity (CDC). This immune effector mechanism is rarely studied.  The authors conducted in vitro studies and discovered that radiotherapy significantly upregulates resistance markers to antibody-based cancer therapy, while also noting that complement activation after radiation enhances the efficacy of complement-dependent cytotoxicity (CDC). The results do agree with their hypothesis, but the mechanism by which these two phenomena of development of radio resistance owing to membrane-bound complement 85 regulatory proteins, or how radiotherapy leads to increased sensitivity to CDC is not explained.

Also, Why the authors have only used in vitro models to study only complement-dependent cytotoxicity while the response to antibody or radiotherapy might involve several immune effector mechanisms, and the effect observed would be combination of all?

Author Response

Revised submission of “Radiotherapy Upregulates the Expression of Membrane-bound Negative Complement Regulator Proteins on Tumor Cells and Limits Complement-mediated Tumor Cell Lysis” (Manuscript ID cancers-3726995)

We greatly appreciate your thorough evaluation of our manuscript and for the constructive suggestions offered. We have revised the paper accordingly and believe that these changes have substantially strengthened the work.

A complete, point-by-point response follows this letter. All text changes are marked in red color. The revised manuscript is submitted as an attachment.

Point-by-Point Response to Reviewer 1:

Comments 1: The authors conducted in vitro studies and discovered that radiotherapy significantly upregulates resistance markers to antibody-based cancer therapy, while also noting that complement activation after radiation enhances the efficacy of complement-dependent cytotoxicity (CDC). The results do agree with their hypothesis, but the mechanism by which these two phenomena of development of radio resistance owing to membrane-bound complement 85 regulatory proteins, or how radiotherapy leads to increased sensitivity to CDC is not explained.

Response 1:

We greatly appreciate the reviewer’s thoughtful remarks. Complement-dependent cytotoxicity (CDC)—the principal effector mechanism for several therapeutic antibodies (e.g., rituximab), but the interplay between ionizing radiotherapy (RT) and the complement system is still only sparsely documented, so we elected to begin with reductionist in-vitro experiments to delineate the underlying cellular and molecular mechanisms. In our study we found that RT may counteract the CDC effect by increasing the expression of mCRPs, providing the innovative insight that the sequence of combined RT and antibody-based immunotherapy should be under consideration and the prior activation of complement system may increase the effect of combined treatment. We fully agree that these findings must ultimately be validated in vivo, where additional innate-immune components and the adaptive immune system will undoubtedly shape the overall response. However, controlling a wide range of potential parameters in vivo would also require many more experiments, which is currently outside our present capacity and scope. Thus our innovative findings in this in vitro study can at least form the foundation of follow-up mechanistic studies and subsequent in-vivo investigations.

Additionally, we have modified and extended our discussion towards how RT leads to increased sensitivity to CDC accordingly. Please see page 13, line 364-372.

Comments 2: Also, Why the authors have only used in vitro models to study only complement-dependent cytotoxicity while the response to antibody or radiotherapy might involve several immune effector mechanisms, and the effect observed would be combination of all?

Response 2: Thank you very much to point this out. In addition to what we have said about potential in vivo data, we do appreciate that this is an important issue and fully recognize that the clinical response to monoclonal antibodies or radiotherapy (RT) reflects an integrated network of effector mechanisms—including antibody-dependent cellular cytotoxicity (ADCC), antibody-dependent cellular phagocytosis (ADCP) etc. We view the current study as the mechanistic “ground floor.” We try to understand how radiation modulates CDC in vitro (e.g., radiation-induced up-regulation of CD46/CD55/CD59/CD20, impact of timing of combined RT and antibody-based immunotherapy) and translate the findings to in vivo experiments. On the other hand, at the start of the project, it was difficult to predict cause-and-effect relationships when multiple effector mechanisms act simultaneously in in-vivo models. We aim to validate our in vitro results in future in vivo experiments with more robust, holistic data.

Reviewer 2 Report

Comments and Suggestions for Authors

Radiotherapy (RT) remains a foundational modality in cancer treatment, known not only for its cytotoxic effects but also for its capacity to modulate immune responses. In their submitted manuscript, the authors explore the complex interplay between RT and the complement system—an essential component of the innate immune defense. This is an important and timely topic, as understanding how RT affects complement activation and regulation could open up novel avenues for combinatorial therapies. The authors demonstrate that RT upregulates membrane-bound complement regulatory proteins (mCRPs), such as CD46, CD55, and CD59, in a dose- and time-dependent manner, which may attenuate complement-mediated tumor cell killing. They also show that complement activation prior to RT enhances early apoptosis in tumor cells. To me, this study offers promising insight into the immunomodulatory effects of RT and suggests rational design of combination strategies involving RT and complement-targeting therapies. I have the following questions/comments:

  1. Figure 1: Please include the histogram plot corresponding to the flow cytometry data in Figure 1 as part of the supplementary figures.
  2. Please describe the gating strategy used in the flow cytometry experiments in the Methods section.
  3. Figure 2: Similarly, please provide the histogram plot of the flow cytometry data shown in Figure 2 in the supplementary figures.
  4. Figure 3: What does “NHS” stand for? Kindly define it in the figure legend.
  5. Regarding CD20 expression: The authors report increased CD20 levels after RT in lymphoma and leukemic cells. Could this have implications for improved sensitivity to anti-CD20 therapies such as rituximab? Please elaborate on how this could be clinically translated.
  6. Complement timing is a key point of interest. Why does complement activation before irradiation enhance apoptosis more effectively than post-RT treatment? Is this related to differences in surface protein availability, membrane integrity, or other factors? Please include mechanistic discussion or speculation.
  7. The authors claim that complement modulation does not influence DNA-damage repair or radiosensitivity. Please include survival curves or additional data from the colony-forming assay to substantiate this conclusion.
  8. The Materials and Methods section needs substantial revision for clarity and reproducibility. Please provide precise experimental protocols including complement source, antibody concentrations, radiation dosages, time intervals, and flow cytometry settings.
  9. Some figure panels require clearer labeling and presentation. Flow cytometry data would benefit from representative histograms or dot plots with well-defined gates. All abbreviations should be spelled out at first use in figure legends.

Comments on the Quality of English Language

N/A

Author Response

Revised submission of “Radiotherapy Upregulates the Expression of Membrane-bound Negative Complement Regulator Proteins on Tumor Cells and Limits Complement-mediated Tumor Cell Lysis” (Manuscript ID: cancers-3726995)

We greatly appreciate your thorough evaluation of our manuscript and for the constructive suggestions offered. We have revised the paper accordingly and believe that these changes have substantially strengthened the work.

A complete, point-by-point response follows this letter. All text changes are marked in red color. The revised manuscript is submitted as an attachment.

Point-by-Point Response to Reviewer 2:

Comments 1: Figure 1: Please include the histogram plot corresponding to the flow cytometry data in Figure 1 as part of the supplementary figures.

Response 1: Thank you for your valuable suggestion. We agree with this comment.

In response to your comment, we have included the representative corresponding histogram plots of cell line Ramos for the flow cytometry data shown in Figure 1 as Supplementary Figure S1A, B. These histograms provide a clearer representation of the fluorescence intensity distributions and support the gating and quantification shown in the main figure. Please see page 15, line 408, supplementary Figure S1.

We appreciate your attention to data presentation and believe this addition improves the clarity and completeness of our results.

Comments 2: Please describe the gating strategy used in the flow cytometry experiments in the Methods section

Response 2: Thank you for your insightful comment. We appreciate your suggestion to clarify the gating strategy used in our flow cytometry experiments. In response, we have revised the Methods section to include a detailed description of the gating strategy. Specifically, we first excluded debris based on forward and side scatter (FSC/SSC) parameters, followed by exclusion of doublets based on forward scatter (FSC-A/FSC-H) parameters. Therefore, we have added this description in the Method section and representative gating strategies in the supplementary figures as you suggested. Please see page 4, line 150-153; page 15, line 408, supplementary Figure S1; page 18, line 437, supplementary Figure S4.

Comments 3: Figure 2: Similarly, please provide the histogram plot of the flow cytometry data shown in Figure 2 in the supplementary figures.

Response 3: Thank you for the instructive advice. We have accordingly provided the corresponding histogram plots of flow cytometry data shown in Figure 2 and named it as Figure S2. Please see page 16, line 423, supplementary Figure S2.

Comments 4: Figure 3: What does “NHS” stand for? Kindly define it in the figure legend.

Response 4: Thank you for your comment. We fully agree that it should be defined properly in the figure legend. NHS was defined in the main text of the manuscript at its first occurrence (see page 3, line 121-125), and it means normal human serum containing the full complement factors that are required for a complete complement cascade.

Therefore, we have accordingly defined in the figure legend. Please see page 10, line 264-265, 268-269.

Comments 5: Regarding CD20 expression: The authors report increased CD20 levels after RT in lymphoma and leukemic cells. Could this have implications for improved sensitivity to anti-CD20 therapies such as rituximab? Please elaborate on how this could be clinically translated.

Response 5: Thank you for your comment. We assume that the enhanced CD20 expression upon radiation (RT) implies a potential synergistic target for the classical anti-CD20 monoclonal antibody treatment (namely rituximab) in B-cell hematological malignancies before RT. On the other hand, according to our observations that the elevation of membrane-bound complement regulatory proteins (mCRPs) after radiotherapy might counteract the cytotoxicity of anti-CD20 treatment because the anti-CD20 treatment relies on complement-mediated cytotoxicity. Thus, our findings indicate that applying anti-CD20 treatment prior to RT would be critical and beneficial for the further clinical translation of this combinatory radio-immunotherapy.

Comments 6: Complement timing is a key point of interest. Why does complement activation before irradiation enhance apoptosis more effectively than post-RT treatment? Is this related to differences in surface protein availability, membrane integrity, or other factors? Please include mechanistic discussion or speculation.

Response 6: We thank the reviewer for this insightful question. We mainly share similar speculations with the reviewer. As it is discussed in our study (please see page 13, line 359-364, line 369-372) that ionizing radiotherapy (RT) up-regulated CD46, CD55 and CD59 (Figure 1 and Figure S1), the regulators accelerating C3b decay and preventing MAC insertion, limiting complement activity if it is delivered after RT. Furthermore, if we activate the complement system before the RT, C3b/iC3b fragments [1] and sub-lytic membrane-attack complexes (MAC) [2] are deposited on the tumor-cell surface while the plasma membrane is still intact without being destroyed by the ionizing RT. This “priming” event by the prior complement activation may lower the threshold of apoptosis (Figure 3) by maintaining the docking sites to elicit the following cell death events.

We hereby extend our discussion accordingly to explain the phenomenon with potential mechanisms. Please see page 13, line 364-369.

Comments 7: The authors claim that complement modulation does not influence DNA-damage repair or radiosensitivity. Please include survival curves or additional data from the colony-forming assay to substantiate this conclusion.

Response 7: Thank you for your comment. We are sorry that previously we put the related graphs in the Appendix A2 and A3 of the manuscript. But considering the accessibility of the figures, we have changed it from Appendix to Supplementary Figure S3 and S4. Please see page 17 and 18.

Comments 8: The Materials and Methods section needs substantial revision for clarity and reproducibility. Please provide precise experimental protocols including complement source, antibody concentrations, radiation dosages, time intervals, and flow cytometry settings.

Response 8: We thank the reviewer for highlighting the need for more methodological detail. We have now expanded and reorganized the Materials and Methods section to facilitate full clarity and reproducibility. Key additions are reflected in the marked-up manuscript. Please check page 3, line 121-125(complement source); page 3-4, line 137-139(antibody concentrations); page 4, line 140-141 (radiation dosages); page 4, line 141 (time intervals) and page 4, line 150-153; page 4, line 178-180, page 5, line 190-193 (flow cytometry settings).

Comments 9: Some figure panels require clearer labeling and presentation. Flow cytometry data would benefit from representative histograms or dot plots with well-defined gates. All abbreviations should be spelled out at first use in figure legends.

Response 9: Thank you for your constructive suggestion. We totally agree with the reviewer and therefore have added the histogram plots and dot plots with well-defined gates in the supplementary figures as you advice. Please check page 15, line 408, supplementary Figure S1; page 16, line 423, supplementary Figure S2; page 18, line 437, supplementary Figure S4.

References:

  1. Zarantonello, A.; Revel, M.; Grunenwald, A.; Roumenina, L.T. C3-dependent effector functions of complement. Immunol Rev 2023, 313, 120-138, doi:10.1111/imr.13147.
  2. Xie, C.B.; Jane-Wit, D.; Pober, J.S. Complement Membrane Attack Complex: New Roles, Mechanisms of Action, and Therapeutic Targets. Am J Pathol 2020, 190, 1138-1150, doi:10.1016/j.ajpath.2020.02.006.

Reviewer 3 Report

Comments and Suggestions for Authors

The manuscript investigates how ionizing radiation (RT) impacts the complement axis in cancer. This study to elucidate the complex intersection between RT and innate immunity is novel and well-designed/carried out. I would recommend acceptance upon minor improvements. 

  1. Discuss further on the plausible mechanisms for the up-regulation of the membrane-bound complement regulators (CD46, CD55, CD59). Any other factors beside C3a/C5a?
  2. Could discuss on the clinical sequencing practicality, on timing safety, comparison with existing RT-immunotherapy protocols.  
  3. Re-analyze dose-response data with one-way ANOVA + post-hoc correction.
  4. Include Appendix A in a separate file and add FACS analysis, gating strategies etc.

Author Response

Revised submission of “Radiotherapy Upregulates the Expression of Membrane-bound Negative Complement Regulator Proteins on Tumor Cells and Limits Complement-mediated Tumor Cell Lysis” (Manuscript ID: cancers-3726995)

We greatly appreciate your thorough evaluation of our manuscript and for the constructive suggestions offered. We have revised the paper accordingly and believe that these changes have substantially strengthened the work.

A complete, point-by-point response follows this letter. All text changes are marked in red color. The revised manuscript is submitted as an attachment.

Point-by-Point Response to Reviewer 3:

Comments 1: Discuss further on the plausible mechanisms for the up-regulation of the membrane-bound complement regulators (CD46, CD55, CD59). Any other factors beside C3a/C5a?

Response 1: Thank you very much for helping us emphasize this very important point in our discussion. Indeed, previous studies observed a cytokine-mediated up-regulation of CD55 and CD59 expression in human hepatoma cells in tumor mediated via pro-inflammatory cytokines such as tumor necrosis factor-alpha, IL-1beta, and IL-6 [1]. Since RT promotes inflammation in cancer by inducing an upregulation of proinflammatory cytokines expression in cancer cells among others [2], RT-induced cytokine induction leading to mCRP upregulation might be another potential mechanism to explore further. Thus, more studies on investigating the interaction between complement system, radiotherapy and cytokine production are required. Accordingly, we have widened our discussion at this point. Please see page 12, line 332-339.

Comments 2: Could discuss on the clinical sequencing practicality, on timing safety, comparison with existing RT-immunotherapy protocols.

Response 2: Thanks for putting forward this interesting and practical question. Among current RT- immunotherapy protocols, such as RT+ anti-PD-1(Programmed Cell Death Protein 1) /PD-L1 (Programmed Death-Ligand 1), RT + anti-CTLA-4 (Cytotoxic T-Lymphocyte-Associated Protein 4), RT + STING (stimulator of interferon genes) agonist, RT is usually followed by immunotherapy or concurrently given with immunotherapy to harness human immune system after RT-induced antigen release or double-strand breaks. A prominent clinical example is here the PACIFIC series in NSCLC (or Adriatic in SCLC) where Durvalumab (PD-L1 inhibitor) after concurrent chemoradiotherapy (cCRT) has evolved as the standard of care for patients with unresectable, stage III non-small-cell lung cancer (NSCLC). This sequence seems clinically safe and effective also for the sequential (sCRT) procedure [3]. However, the sequencing of RT-Immunotherapy integration in RT regimens remains a matter of debate, as concurrent schedules are effectively used in many preclinical models. 

Our findings in this study show that RT should be performed after complement-activation-based immunotherapy, because RT could increase the expression of radio-resistant markers, mCRPs, and reduce the treatment efficiency of antibody-based immunotherapy. Meanwhile, this strategy may also reduce the severity of RT-enhanced vascular leakage, which improves the treatment safety. We have addressed on this clinical sequencing practicality. Please see page 13, line 372-385. Thus the potential sequence suggestion might be: complement-activation-based immunotherapy followed by RT (+/- chemotherapy) followed by checkpoint inhibition.

Comments 3: Re-analyze dose-response data with one-way ANOVA + post-hoc correction.

Response 3: Thank you very much for pointing this out! We have re-analyzed all dose response data with one-way ANOVA respectively and subsequent correction for multiple comparisons with Dunnett’s multiple comparisons testing chosen as recommended by GraphPad Prism version 10 for the corresponding data structure.

  • For Figure 1, please compare figure and figure legend, page 6, line 216, 224: Data analysis was performed using one-way ANOVA with post-hoc Dunnett’s multiple comparisons testing.
  • For Figure S1, please compare figure and figure legend, page 15, line 417: Data analysis was performed using one-way ANOVA with post-hoc Dunnett’s multiple comparisons testing.
  • For Figure 2, please compare figure and figure legend, page 8, line 238; page 9, 242-243: Data analysis was performed using one-way ANOVA with post-hoc Dunnett’s multiple comparisons testing.

Comments 4: Include Appendix A in a separate file and add FACS analysis, gating strategies etc.

Response 4: Thank you very much for helping us improve our data presentation. We have now included our gating strategies and representative histogram plots of the FACS analysis into our supplementary figures, respectively. Please check page 15, line 408, supplementary Figure S1; page 16, line 423, supplementary Figure S2; page 18, line 437, supplementary Figure S4.

References:

1.         Spiller, O.B.; Criado-García, O.; Rodríguez De Córdoba, S.; Morgan, B.P. Cytokine-mediated up-regulation of CD55 and CD59 protects human hepatoma cells from complement attack. Clin Exp Immunol 2000, 121, 234-241, doi:10.1046/j.1365-2249.2000.01305.x.

2.         Schaue, D.; Micewicz, E.D.; Ratikan, J.A.; Xie, M.W.; Cheng, G.; McBride, W.H. Radiation and inflammation. Semin Radiat Oncol 2015, 25, 4-10, doi:10.1016/j.semradonc.2014.07.007.

3.         Garassino, M.C.; Khalifa, J.; Reck, M.; Chouaid, C.; Bischoff, H.; Reinmuth, N.; Cove-Smith, L.; Mansy, T.; Cortinovis, D.L.; Migliorino, M.R.; et al. Durvalumab after sequential chemoradiotherapy in unresectable stage III non-small-cell lung cancer-final analysis from the phase II PACIFIC-6 trial. ESMO Open 2025, 10, 105071, doi:10.1016/j.esmoop.2025.105071.

Round 2

Reviewer 2 Report

Comments and Suggestions for Authors

I noticed that Figure 1 in the revised PDF version lacks labeling, which seems to be an inadvertent error. Kindly revise it accordingly.

Author Response

Revised submission of “Radiotherapy Upregulates the Expression of Membrane-bound Negative Complement Regulator Proteins on Tumor Cells and Limits Complement-mediated Tumor Cell Lysis” (Manuscript ID: cancers-3726995)

Thank you once again for your thoughtful and thorough evaluation of our manuscript. We have carefully addressed the comments and revised the paper accordingly.

A detailed, point-by-point response is provided below. All changes made in the manuscript are highlighted in red. The revised version has been submitted as an attachment.

Comment 1:
I noticed that Figure 1 in the revised PDF version lacks labeling, which seems to be an inadvertent error. Kindly revise it accordingly.

Response 1:
Thank you for pointing out this oversight. The missing labels resulted from the embedding of Figure 1 as a PDF into the Word document, which inadvertently led to formatting loss. We have now corrected this issue and also incorporated suggestions from the academic editor. The revised and fully labeled version of Figure 1 is available on page 6, line 219 of the updated manuscript.